# Continuous then discrete: A recommendation for building robotic brains

**Chris Eliasmith**
Centre for Theoretical Neuroscience
University of Waterloo
Canada
celiasmith@uwaterloo.ca

**P. Michael Furlong**
Centre for Theoretical Neuroscience
University of Waterloo
Canada
michael.furlong@uwaterloo.ca

**Abstract:** Modern neural networks have allowed substantial advances in robotics, but these algorithms make implicit assumptions about the discretization of time. In this document we argue that there are benefits to be gained, especially in robotics, by designing learning algorithms that exist in continuous time, as well as state, and only later discretizing the algorithms for implementation on traditional computing models, or mapping them directly onto analog hardware. We survey four arguments to support this approach: That continuum representations provide a unified theory of functions for robotic systems; That many algorithms formulated as temporally continuous demonstrate anytime properties; That we can exploit temporal sparsity to effect energy efficiency in both traditional and analog hardware; and that these algorithms reflect the instantiations of intelligence that have evolved in organisms. Further, we present learning algorithms that are derived from continuous representations. Finally, we discuss robotic precedents for this approach, and conclude with the implications of using continuum representations in robotic systems.

**Keywords:** Continuous representations, Neuromorphics, Learning

## 1 Introduction

Neural networks have enabled profound advances in robotic capabilities, but carry unexamined assumptions about the discretization of time. Robots are embedded in the physical world and in real time. Making 'atemporal' algorithms fast enough to behave responsively comes at considerable costs of power, volume of computing equipment, and engineering time – limiting what can be fielded in real robots. Just as robotics has benefitted from modelling control problems in the continuum before implementing discretized algorithms, we argue that machine learning approaches should be derived from temporally continuous models first, and subsequently discretized for implementation as needed.

In fact, roboticists are in a unique position to appreciate this argument. Robots are physical systems interacting with physical environments operating in continuous time. Our relevant physical theories for robotics are all continuous. Continuous characterizations of the brains that drive those physical systems provides a unity of approach not otherwise achievable, and may yield efficiencies that might not otherwise be obvious.

Specifically, we propose that the design and characterization of machine learning algorithms should begin with assumptions that they are operating in continuous time. It follows then that the best mathematical methods for capturing the computations being performed will typically rely on dynamic systems theory and related continuous approaches (e.g., control theory, functional analysis). Currently, such systems must be approximated by discrete computing hardware for implementation purposes, but starting with continuity and ending with discreteness is quite different from the approach typically taken in machine learning and AI.

Contemporary AI approaches, both neural network and symbolic methods, tend to assume temporal discreteness and hope that the speed of computation (or reduction in step size) will make up for

Blue Sky Papers, 5th Conference on Robot Learning (CoRL 2021), London, UK.

the fact that the real-world is temporally continuous. This is true for both recurrent and feedforward neural networks. One fundamental manifestation of temporal discreteness is that a standard artificial 'neuron' lacks temporal properties: it is purely a function of the input state, not of time. This subordinates a system's dynamics to the network architecture, ignoring any relation between real time and computation time. But, time necessarily passes between and during layers of computation. The standard assumption, that nothing changes in the real time between two computational steps, is appropriate only if your time steps are small compared to the dynamics of the system you are interacting with. This is often not the case. And, perhaps worse, it can be extremely energetically expensive to make your system run fast enough to 'ignore' time.

In this brief paper, we canvass four arguments for why a continuity-first approach should be embraced within robotics for building algorithms to run robust, adaptive systems. We then discuss several consequences of adopting this view, and briefly survey some recent techniques that support it, highlighting their proven advantages. Ignoring continuous-time machine learning may limit the performance and responsiveness of algorithms, while increasing power demands, severely constraining the ability of future robots to learn.

## 2 Four Arguments

### 2.1 The unification argument

Our best theories of phenomena like control and sensing start with continuity in state and time (e.g., modern linear or nonlinear control theory and signal processing). There are many discretized implementations of such algorithms, and it is fair to say that recent neural networks for many control and sensing problems tend to be discretized over time. However, returning to initial continuous characterizations can provide insights not otherwise evident (see section 3 for some examples). Notably, many of our best neural networks do start with continuity assumptions with respect to state. For instance, CNNs, a sensory processing powerhouse, are based on convolutions. Furthermore, our best algorithms for higher-level phenomena — like language processing [1] and abstract game playing [2] — also begin by considering conceptual representations as continuous vectors in a high-dimensional space.

The Hamilton-Jacobi-Bellman equations [3] unify continuous-time optimal control with reinforcement learning (RL) [4, 5]. Doya suggests that continuous-time RL abrogates the complications of improper discretization - too coarse and the control is poor, too fine and the state space explodes. Recently, Lutter *et al.* [6, 7] have constructed a robust Value Iteration algorithm, that demonstrates robustness to the simulation to reality transition, a benefit, they infer, due to the continuous time representation.

Consequently, both sensory and cognitive models would do well to embrace continuity over both space and time. These observations suggest that a unified theory of the kinds of functions we are interested in having in our robots can best be described in terms of the evolution of continuous vector spaces over continuous time.

### 2.2 The anytime argument

Recent work in computer science has found what are often termed 'anytime' algorithms to be especially useful for flexible resource allocation, and intelligent systems more generally [8]. This class of algorithm provides a result whenever they are interrupted, and the later they are interrupted, the better the result. Unsurprisingly, such features are valuable in the context of real-time, real-world robotics. Dynamic, ever changing environmental demands means that such systems often cannot determine how much time is available for arriving at an answer to a given problem. Thus a continuously improving result will be as good as it can be for the allotted resources.

It is the authors' experience that constructing algorithms in continuous space and time encourages the discovery of these kinds of algorithms. When neural network components operate continuously, including in time, anytime algorithms become more prominent, if for no other reason than the networks must evolve towards the desired output. While it is not strictly the case that any continuous algorithm is 'anytime', it is true that many algorithms formulated in this manner (including all of those mentioned in section 3) provide online, real-time results and often, if not always, with im-

provement as time progresses. Interestingly, in the case of spiking neural networks, even the basic method of estimating the value of a transmitted signal has these properties [9].

## 2.3 The evolutionary argument

Traditional AI prioritizes reasoning using discrete symbols and discrete time. Modern neural network approaches to classification and object identification often make the states (effectively) continuous, but seldom consider continuous time. However, in nature non-symbolic and symbolic cognition run on a continuous, nonlinear, dynamical computing substrate. Animals remain the prime example of robust, multi-function, intelligent systems. Indeed, the desire to attain animal-like efficiencies in computing is the animating force behind the early neuromorphic computing research programme [10].

One might counter that biological neurons communicate discretely using spikes, however the picture is more complex. First, neural spikes occur at continuous moments in time, and that timing can be critical to effective neural processing [11]. Second, Spikes predominantly occur in long distance communication, e.g., between brain regions, and not over short distances, such as in the retina [12]. This suggests discrete spikes enable communication of continuous state variables over long distances in noisy environments [9].

The approach that evolved in nature was to start with continuous computations, and only pay the cost for discrete mechanisms when efficiency demands. That evolutionary processes produced this solution suggests that characterizing algorithms in terms of temporally discrete computing in robots may be imposing a form of computation that misses efficiencies of operating in continuous time.

## 2.4 The energy efficiency argument

Energy efficiency is critical to deploying autonomous systems with significant amounts of intelligence. Without improvements to on-board energy density, gains must be realized in energy use. Adopting a continuity-first approach can greatly improve energy efficiency as it provides for innovations that otherwise may go unnoticed. For instance, characterizing temporal computations as continuous and introducing dynamics into single neurons can result in very temporally sparse computations that perform as well as their dense counterparts [13]. In short, thinking carefully about continuous dynamics when processing temporal signals leads to an analogous 'sparsification' to that commonly used in connection weight matrices in contemporary ANNs.

Perhaps the most exciting connection between continuity and efficiency can be found in analog computing substrates, which are extremely efficient, but often difficult to program. Recent suggestions for analog neuromorphic devices are built with a continuity-first approach at the core [14], characterizing their programming in terms of continuous variables over state and time. The extremely low-power, but generalizable computing achieved by this work [15] bolsters the notion that our greatest efficiencies will likely be achieved from a continuity-first perspective.

## 3 Current methods that are continuity-first

The Neural Engineering Framework [NEF; 9] provides three principles for implementing nonlinear dynamics in spiking (or non-spiking) neural networks. It assumes both continuous state and time. The Semantic Pointer Architecture [SPA; 16] is built on top of the NEF to provide an architecture based on the mammalian brain that is used to build cognitive systems broadly construed, and underwrites the world's largest functional brain model, Spaun [17, 18]. The SPA provides continuous characterization of symbolic and sub-symbolic processing. These are both broad, general frameworks that adopt a continuity-first approach.

There are more specific examples of the benefits of beginning with continuous temporal algorithms as well. For instance, the Legendre Memory Unit [LMU; 19], an RNN that has recently been used to beat a wide variety of SoTA benchmarks, begins by discovering an optimal continuous dynamical system for temporal compression [20, 21]. To be clear, the original formulation of the core of the LMU is as a continuous time dynamical system, which is subsequently discretized in a variety of manners (e.g., using ZOH, or Euler's method). The discretized system is then run on standard ML benchmarks. Beginning with a continuous time dynamical system has provided the LMU with a

significantly improved ability to capture long term dependencies, partly because it is formulated independently of a notion of a 'time step'.

Similarly, the REACH adaptive controller [22] is a continuous time, online, adaptive controller that constantly tunes a control system, in a manner inspired by the mammalian motor system. The formulation of this controller as continuous in time both for execution and learning allows it to be discretized in a manner that can exploit the efficiencies of neuromorphic hardware. As a result, REACH has been implemented on Intel's recent Loihi neuromorphic hardware to achieve state-of-the-art efficiency and accuracy on a robot [23]. This method is notable for not only starting with continuity, but incorporating continuous learning while maintaining performance guarantees.

## 4  Discussion

Roboticists have heard suggestions similar in some senses to this one before. Potential fields methods [24] and Brooks' subsumption architecture [25] are highly focused on dynamical systems approaches to robotics. However, subsumption does not specifically address the importance of continuity, but rather proposes a means of arranging interacting components. Regrettably, that specific architecture falls short when it comes to larger, more sophisticated, and more intelligent systems. Some of the recent developments, discussed above (e.g., the SPA), are aimed at overcoming those concerns, but in many ways solutions to the coordination problem are independent of the claims we are making. Our focus is on the advantages — efficiency, algorithmic, and theoretical unity — of starting algorithmic design from the perspective of continuous time (and state).

It is helpful to reiterate some consequences of adopting a continuity-first view. One clear practical consequence is that significant benefits are likely to come from more carefully considering hardware substrates. Specifically, including dynamics in the neurons (or computational components more generally) used to build intelligent systems should be a priority. Having hardware constraints match algorithmic assumptions, instead of the reverse, generally leads to efficient implementations. Analyzing algorithms abstracted from particular hardware provides the space to recognize those assumptions, and identify better computing substrates. Such an approach has been widely embraced by neuromorphic hardware developers, but there may be other ways in which dynamics can permeate hardware design to improve efficiency. Robots are inherently hybrid discrete/continuous systems, operating in continuous environments in continuous time. Discrete computing has provided substantial benefits in terms of reliability, as well as flexibility and ease of programming. But the benefits of discrete representations come at a price — power, computation time — that may not be compatible with fielding responsive, capable, intelligent systems.

Further, discrete computation comes with built-in assumptions about precision and temporal resolution that may not be matched to the problem. By reducing the use of discrete representations in a robotic system's algorithmic design we forestall the number of times these assumptions must be made, ideally leaving them until we are at the hardware level. Perhaps the development of analog hardware will one day remove the requirement to make them at all [14, 26].

In conclusion, we believe that each and all of the arguments presented, in combination with specific example methods that have recently been developed make a strong case for a continuity-first approach to adaptive robotics. While our considerations here have been brief, there is a broad and growing literature that embraces this perspective. We hope that this paper serves as an introduction to some of the benefits of starting with continuity, and if nothing else, provokes a discussion on the topic of typical current assumptions when building intelligent robotic systems.

**Acknowledgments**

This work was supported by CFI and OIT infrastructure funding as well as the Canada Research Chairs program, NSERC Discovery grant 261453, NUCC NRC File A-0028850.

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
