# OpenReview forum: "Continuous then discrete: A recommendation for building robotic brains"
_robot-learning.org/CoRL/2021/Conference/Blue_Sky — CoRL 2021, Blue Sky_

### Official Review · Reviewer_Rjxo · 2021-08-24

**Novelty:** Good
**Impact:** 4
**Clarity Of Presentation:** Very Good

**Recommendation:**

Strong Accept: I recommend accepting the paper and will argue for my recommendation even if other reviewers hold a different opinion.

**Summary:**

This paper advocates for the position that we should think about robot learning architectures and algorithms first and foremost from a perspective of continuous time. The paper argues that four important points motivate this need (1) unification; (2) anytime algorithms; (3) evolution; and (4) energy efficiency.

The authors highlight a number of recent works which have benefited from this continuous time viewpoint and have an impassioned call that others in the field should take up the cause of continuous time algorithms for robot learning. The paper further discusses that even when using discrete algorithms or hardware, thinking about continuous time first can improve our understanding of the problem as well as developing better algorithms.

I am convinced by the paper's argument that we should look more at continuous time algorithms in robot learning.

I would point the authors to a few recent works on continuous time value iteration methods which have strong implications for robot reinforcement learning and further support the arguments in this paper:
 - Lutter, M.; Mannor, S.; Peters, J.; Fox, D.; Garg, A. (2021). Robust Value Iteration for Continuous Control Tasks, Robotics: Science and Systems (RSS).  https://arxiv.org/pdf/2105.12189.pdf
- Lutter, M.; Mannor, S.; Peters, J.; Fox, D.; Garg, A. (2021). Value Iteration in Continuous Actions, States and Time, International Conference on Machine Learning (ICML).   https://arxiv.org/pdf/2105.04682.pdf



**Summary Of Recommendation:**

I find this paper to be the perfect fit for the blue sky track. I think it provides a good mix of background, argument, and speculation that continuous time representations are worth more focus in our community. I think it would spark interesting discussions at the conference if accepted.

---

### Official Review · Reviewer_JSra · 2021-08-30

**Novelty:** Very Good
**Impact:** 4
**Clarity Of Presentation:** Very Good

**Recommendation:**

Strong Accept: I recommend accepting the paper and will argue for my recommendation even if other reviewers hold a different opinion.

**Summary:**

- Authors question if we aren't giving continuity the due credit it deserves.
- Sound arguments are made on why continuity deserves a second thought.
- Possibly in some of the biological arguments there is an implicit assumption that mimicking biology is beneficial. However, I agree that it can be a good prior/ candidate that needs investigation.



**Summary Of Recommendation:**

After a short while, it's often a good idea to stop and look back. The questions being asked are spot on. It's time to reflect -- are we on the right path, did we miss anything?

---

### Official Review · Reviewer_Mvy3 · 2021-09-03

**Novelty:** Very Good
**Impact:** 2
**Clarity Of Presentation:** Very Good

**Recommendation:**

Weak Reject: I recommend rejecting the paper, but will not argue for my recommendation if the majority of other reviewers have a different opinion.

**Summary:**


This position paper argues that policy learning in robotics should happen in a continuous time (CT) framework. The contention is that most current learning work adopts a discrete time (DT) framework and that this approach precludes algorithms or efficiencies that might otherwise be possible. The paper gives four arguments for this position, perhaps the strongest of which is that biological systems appear to be fundamentally CT.

I thought that the thesis (CT rather than DT) of the paper was interesting but that there wasn't a lot of technical justification for the idea, aside from a neuromorphic justification and energy efficency arguments. Out of the four arguments made, the one that made the most sense to me was the evolutionary argument, which I interpreted as a neuromorphic argument, i.e. we should design CT systems b/c that's how animals function. I also thought that it was reasonable to say that there was the potential for CT based systems to be more energy efficient.

There were a few statements that struck me as out of place:

Line 54: "Our best theories of phenomena like control and sensing start with continuity in state and time (e.g., modern linear or nonlinear control theory and signal processing)." This seems like an odd statement to me. Most control theory can be framed either in CT or DT, e.g. LQR/LQG. In fact, nonlinear control is frequently framed in DT b/c it is solved as a non-linear optimization problem.

Line 74: "Constructing algorithms in continuous space and time encourages the discovery of these kinds of algorithms (i.e. anytime algorithms). It is not at all clear to me why a CT framework makes anytime algorithms easier to develop.

Line 86: "Characterizing algorithms in terms of temporally discrete computing in robots may be imposing a form of computation that misses efficiencies of operating in continuous time." This is an interesting statement, but there's little justification and it seems like speculation.



**Summary Of Recommendation:**

I didn't see a lot of justification here for the high level thesis that continuous time systems are to be preferred over discrete time systems. I also saw several seemly unsupported claims.

---

### Decision · Program_Chairs · 2021-10-01

**Decision:**

Accept

**Comment:**

The paper argues from several viewpoints the importance of continuous time for modeling robot systems. It is expected to spur interesting discussions during the conference. The authors are encouraged to
* tighten up the statements pointed out by Reviewer Mvy3 and make them technically precise
* address the counterfactual point. While the paper is quite right about the importance of continuous, how do we explain the vast success of discrete-time models? When will continuous-time play a CRITICAL role? What is the gap?